# Revealing Hidden Features in Multilayered Artworks by Means of an Epi-Illumination Photoacoustic Imaging System

**DOI:** 10.3390/jimaging7090183

**Published:** 2021-09-10

**Authors:** George J. Tserevelakis, Antonina Chaban, Evgenia Klironomou, Kristalia Melessanaki, Jana Striova, Giannis Zacharakis

**Affiliations:** 1Institute of Electronic Structure and Laser, Foundation for Research and Technology Hellas, GR-70013 Heraklion, Crete, Greece; ph4831@edu.physics.uoc.gr (E.K.); alina@iesl.forth.gr (K.M.); zahari@iesl.forth.gr (G.Z.); 2National Institute of Optics INO-CNR, 50125 Florence, Italy; jana.striova@cnr.it

**Keywords:** photoacoustic, imaging, diagnostics, cultural heritage, artwork, underdrawings

## Abstract

Photoacoustic imaging is a novel, rapidly expanding technique, which has recently found several applications in artwork diagnostics, including the uncovering of hidden layers in paintings and multilayered documents, as well as the thickness measurement of optically turbid paint layers with high accuracy. However, thus far, all the presented photoacoustic-based imaging technologies dedicated to such measurements have been strictly limited to thin objects due to the detection of signals in transmission geometry. Unavoidably, this issue restricts seriously the applicability of the imaging method, hindering investigations over a wide range of cultural heritage objects with diverse geometrical and structural features. Here, we present an epi-illumination photoacoustic apparatus for diagnosis in heritage science, which integrates laser excitation and respective signal detection on one side, aiming to provide universal information in objects of arbitrary thickness and shape. To evaluate the capabilities of the developed system, we imaged thickly painted mock-ups, in an attempt to reveal hidden graphite layers covered by various optically turbid paints, and compared the measurements with standard near-infrared (NIR) imaging. The obtained results prove that photoacoustic signals reveal underlying sketches with up to 8 times improved contrast, thus paving the way for more relevant applications in the field.

## 1. Introduction

In an emerging position among imaging technologies during the last decade, we find photoacoustic (PA) imaging, a novel methodology developed mainly in the context of contemporary biomedical research studies. In this direction, the highly promising capabilities of PA diagnosis have been recently exploited in several applications involving the in vivo acquisition of valuable anatomical, molecular, functional, and flow dynamic information, towards the understanding of fundamental biological mechanisms such as cancer formation and growth [1]. PA imaging is based on the formation of acoustic waves following the absorption of intensity-modulated (typically pulsed) optical radiation by a material [2]. During the incidence of a short light pulse, a portion of the absorbed optical energy is converted into heat, inducing a rapid thermoelastic expansion of the medium and the subsequent generation of an initial pressure that propagates in space in the form of ultrasonic waves. These acoustic waves, typically found in the MHz frequency regime, are usually recorded in time using the same detection equipment (i.e., piezoelectric elements) as in conventional ultrasound imaging. The amplitude of the generated PA waves is proportional to the absorption coefficient of the medium for the employed excitation wavelength [3]. As a result, the PA technique provides optical absorption imaging contrast with 100% relative sensitivity (i.e., a given percentage change in the optical absorption coefficient yields the same percentage change in the PA amplitude) [4].

By selectively transforming optical absorption information into ultrasonic waves characterized by up to three orders of magnitude higher transmissivity, compared to near-infrared (NIR) radiation [5], PA imaging can be used for in-depth investigations of optically turbid media while retaining high spatial resolution, albeit generally lower than pure optical microscopy methods. The trade-off between imaging depth and spatial resolution can be generally tuned according to the detection bandwidth of the employed ultrasonic transducer, thus allowing for the diagnosis of various objects with diverse optical, mechanical, and structural properties [6].

Despite the fact that the vast majority of studies have utilized PA imaging for biomedical applications, recent works have demonstrated the remarkable potential of such technologies in artwork diagnostics. More specifically, different PA-based techniques have been employed for the visualization of hidden underdrawings in paintings [7,8], the uncovering of text in multilayered documents [9], as well as the thickness measurement of thin paint layers through the analysis of PA signal attenuation [10,11]. Nevertheless, all the presented PA diagnostic systems optimized for such applications have been strictly limited to the imaging of relatively thin objects due to the detection of signals in transmission geometry. Therefore, as signal excitation occurs across the lower surface of the artwork, while respective detection at the upper one, the generated ultrasonic waves would heavily attenuate during their propagation inside objects with a thickness of more than a few mm. Unavoidably, this issue restricts seriously the applicability of the demonstrated PA imaging methodologies, hindering investigations over a wide range of cultural heritage (CH) objects with diverse geometrical features.

To overcome such limitations, we present here a novel PA imaging apparatus developed in an epi-illumination geometry, integrating laser excitation and respective ultrasonic detection on one side, aiming to provide universal diagnostic information in objects of arbitrary thickness and shape. The setup was oriented towards diagnostic applications in heritage science and was optimized in terms of excitation, detection, and scanning parameters to ensure a sufficient imaging performance. In this manner, we demonstrate that the effectiveness of PA imaging can complement substantially the existing state of the art methods for this purpose, including visible and NIR optical imaging [12,13,14,15], optical coherence tomography (OCT) [16,17], multiphoton microscopy [18], THz imaging [19,20,21], and X-ray-based techniques [22,23,24,25]. To evaluate the capabilities of the system, we imaged wall painting mock-ups prepared on thick gypsum-based substrates, in an attempt to reveal hidden graphite layers covered by various optically turbid paints. We additionally investigated the PA imaging performance in terms of acquired underdrawing’s contrast as a function of overlying the paint layer’s thickness and performed a relative comparison with a standard optical method, highlighting further the capabilities of the developed system.

## 2. Materials and Methods

### 2.1. PA Imaging Setup

The imaging apparatus (Figure 1a) employed a Q-switched Nd:YAG laser (SL404, Spectron Laser Systems, Rugby, UK; maximum pulse energy 30 mJ, pulse duration: 10 ns, pulse repetition rate: 10 Hz) emitting infrared radiation at 1064 nm for the efficient excitation of PA signals. The beam was initially attenuated and reduced in diameter to 1.2 mm using an adjustable iris diaphragm so that the pulse energy at the sample’s plane is less than 1.73 mJ. A positive lens with a focal distance equal to 50 cm was used to focus loosely (spot size: ~1 mm) the optical radiation, improving the sensitivity of the imaging system. Each sample was placed at the bottom of a 3D-printed sample holder filled with distilled water, which served as an immersion medium for the efficient propagation and subsequent detection of PA signals. The sample’s front surface was irradiated to generate laser-induced ultrasound from the underlying hidden sketch regions. The generated PA waves were transmitted through the paint layer and water prior to their detection in reflectance configuration by a broadband, spherically focused piezoelectric transducer (HFM28, SONAXIS, Besancon, France; nominal central frequency 73 MHz; focal distance: 4.53 mm; numerical aperture: 0.44). The signals were subsequently enhanced by two low-noise radio frequency (RF) amplifiers (TB-414-8A+, Mini-Circuits, Camberley, UK; gain: 31 dB) connected in series to achieve a total gain of 62 dB, which was adequate for the digitization and recording of PA waveforms by an oscilloscope (DSO7034A, Agilent Technologies, Santa Clara, CA, USA; bandwidth: 350 MHz; sample rate: 2 GSa/s). To form an image, the painting was raster scanned with high-precision XY motorized stages (8MTF-75LS05, Standa, Vilinius, Lithuania), to attain a point-by-point data acquisition synchronized with the trigger signal of the laser source. The recorded waveforms (Figure 1b) were averaged two times for signal-to-noise ratio (SNR) improvement, transferred to a computer, and bandpassed between 100 kHz and 30 MHz for high-frequency noise elimination before the estimation of the peak-to-peak PA amplitude value providing the contrast of the resulting 8 bit images. Depending on the size of the underlying sketch area, the scanning regions had dimensions ranging between 2 × 2 to 4.5 × 4.5 cm^2^, respectively, and were sampled, in all cases, using a pixel size of 300 × 300 μm^2^. The total time required for the recording of a PA image ranged from 2.5 to 4 h. Control and synchronization of the PA imaging system were accomplished using custom-developed software, whereas image processing was performed through ImageJ and MATLAB programming environment.

### 2.2. Sample Preparation Procedures

Various geometric patterns and drawings were produced on the prepared substrate composed of gypsum, when it was completely dry, using a graphite pencil (CASTELL B, Faber-Castell, Stein, Germany), representing the hidden underdrawings of the artwork. Subsequently, four characteristic types of pigments—namely, ultramarine blue (Na_7_Al_6_Si_6_O_24_S_3_, Kremer 4503), chromium green (Cr_2_O_3_, Kremer 44200), minium (Pb_3_O_4_, Kremer 42500), and titanium white (TiO_2_, Kremer 46200), were mixed with an acrylic binder (Lascaux Acrylic Adhesive 498 HV) to form thick acrylic paints. In order to produce samples of various paint thicknesses, frames with an increasing number of successive layers of tape (scotch-magic tape) were generated on top of the gypsum substrates. Various mixtures of paints were applied with a spatula, using the tape frames as a guide, over the pencil sketches, forming paint layers with a controlled thickness ranging between 200 and 300 μm.

### 2.3. Analog Profilometer

The thickness of the applied paint layers on the gypsum substrates was measured with an analog profilometer (Perthometer S5P, Mahr, Göttingen, Germany). The stylus of the profilometer is free to move in the vertical axis and is dragged horizontally by the tracer along the measuring path on the surface of the sample. The movement of the stylus tip is subsequently transformed into voltage values portraying the traced profile. The tracer is moved by a drive unit while maintaining a constant speed along the measured path over the surface of interest. An amplifier achieves the necessary vertical magnification of the traced profile. The results for all surface parameters were determined by a microcomputer during the tracing process. The traced profile was automatically printed on millimeter paper, which was subsequently digitized by a photographic camera and processed using custom-developed MATLAB scripts.

### 2.4. NIR Imaging

Images in the NIR were recorded using a custom multispectral imaging system dedicated to paintings diagnostics. The system employs a high-resolution CMOS camera (UI-5480CP, IDS, Obersulm, Germany; 4.92 MP), coupled with a NIR-transparent objective lens (Macro Lens 25 mm F 1.3, Electrophysics Corp., West Fairfield, NJ, USA), whereas the respective illumination is achieved using two broadband emission lamps (Halostar Starlite, OSRAM, Munich, Germany; 50 W, 12 V) placed at 45°. For underdrawings detection, images were collected through a bandpass filter (1200BP25, Omega Optical, Brattleboro, VT, USA; central wavelength: 1100 nm; bandwidth: 25 nm) with the camera positioned at 0°. Angles were expressed with respect to the axis orthogonal to the sample surface. The typical pixel size for the recorded NIR images was equal to 20 μm.

## 3. Results

### 3.1. Initial Performance Evaluation of the PA Imaging System

As a first step, we investigated the performance of the developed reflection-mode PA imaging system as regards the specific visualization of hidden underdrawings and compared the obtained results with pure optical imaging (Figure 1c). For this purpose, a round mock-up sample with a diameter of 4 cm was imaged aiming to reveal an underlying pencil sketch that was covered by a ~200 μm thick paint layer consisting of titanium white, gypsum, and ultramarine blue pigment. Figure 2a depicts an image of the prepared sample, whereas Figure 2b corresponds to the initial pencil sketch prior to the application of the paint layer. A maximum amplitude projection (MAP) PA reconstruction of the sample is shown in Figure 2c, revealing clearly the underdrawing with high specificity resulting from the significantly higher absorption of the incident NIR radiation by the graphite deposition regions, compared to both the gypsum substrate and the overlying layer. The lateral resolution of the recorded PA image is adequate for the delineation of the majority of underdrawing’s spatial details, despite the fact that pencil lines appear slightly thicker, compared to the original sketch presented in Figure 2b. This observation can be explained by considering the finite pixel size of 300 μm that has been used for sampling, providing an image resolution of approximately 600 μm according to Nyquist’s criterion. It has to be mentioned, however, that the resolution limit of the PA imaging setup is ultimately determined by the transducer’s focus, which for the employed detector can be theoretically estimated at 33 μm. Furthermore, a NIR image of the sample was recorded using an excitation spectral band similar to the PA, enabling thus a direct performance comparison between the two imaging techniques. As demonstrated by Figure 2d, the respective NIR image reveals the hidden sketch with similar resolution but a substantially reduced imaging contrast when compared to the PA image of Figure 2c. In this case, the superiority of the PA reconstruction over the pure optical imaging technique can be mainly attributed to the single pass of optical radiation and the dramatically higher transmissivity of the generated PA signals relative to light.

### 3.2. PA Imaging of Underdrawings Covered by Different Pigmented Layers

Having performed an initial validation on the detection capabilities of the developed PA system, we proceeded to the imaging of mock-ups covered by different pigmented layers with an average thickness approximating 250 μm. This study intended to evaluate how different types of NIR transparent pigments can influence the acquired PA contrast, performing additionally a relative comparison with the standard optical technique. Within this framework, we generated four samples whose underdrawings were covered by paint layers containing titanium white, minium, and ultramarine blue pigments. Figure 3a depicts a mock-up covered with a titanium white paint layer, whereas Figure 3b shows the underlying pencil sketch before the application of the paint. A MAP PA image of the sample covering an area of 2 by 2 cm^2^ is explicitly presented in Figure 3c to reveal the hidden “R” letter with high contrast in relation to the background. A NIR optical image at 1100 nm is further shown in Figure 3d, uncovering also the underdrawing albeit with lower imaging contrast, in correspondence to the case presented in Figure 2d. Respective series of images are demonstrated for samples covered by minium (Figure 3e–h), minium and titanium white mixed in equal proportion (Figure 3i–l), as well as ultramarine blue and titanium white mixed in a 14–86% weight ratio (Figure 3m–p) paint layers, providing comparable results in terms of a qualitative underdrawing’s visualization.

Aiming to quantify and compare the imaging performance between the PA and the NIR optical techniques, we have selected, among various image quality metrics [26,27], to estimate the respective Michelson contrast values, which have been representative of the visibility of the hidden underdrawings. To this end, we initially contrast stretched all the images by saturating 0.3% of the pixels and selected five representative pixel brightness profiles vertically to the sketch lines. For each measured profile, the contrast value ***C*** was estimated according to the following relation:$$C=\frac{A_{line}-A_{backg}}{A_{line}+A_{backg}}$$
where ***A_line_*** and ***A_backg_*** correspond to the average brightness value of the pixels representing, respectively, the graphite sketch line and the gypsum sample background. The final contrast for each image was estimated by taking the average out of the five selected profiles, to compensate for potential local signal variabilities.

The contrast quantification results are presented in Table 1, demonstrating an up to 6.8 times higher performance of the PA technique in comparison to the conventional NIR optical imaging. Furthermore, it is observed that PA contrast presents a very limited variability among the four different samples (coefficient of variation—CV% = 6.8%), making the technique virtually independent of the examined pigmented layer type. On the contrary, NIR imaging presents more than 4.8 times higher contrast variability among the samples (CV% = 33%), which reveals its strong dependence on the individual optical absorption and scattering properties of each pigment. The universal imaging capabilities provided by the PA technique can be predominantly attributed to the fact that the possible attenuation of the propagating ultrasonic waves is related to the mechanical and thermodynamic behavior of the paint layer (determined mainly by the acrylic binder), rather than to its optical properties. Nevertheless, we have to clarify that the optical properties of the paint layer may affect significantly the contrast of the recorded PA image, especially when the optical absorption of the pigment becomes comparable to the graphite comprising the underdrawings for the selected excitation wavelength.

### 3.3. Evaluation of PA Contrast as a Function of Layer Thickness

Aiming to explore the effect of the overlying paint layer thickness on the contrast of PA images, we generated five mock-up samples covered with paint layers containing a mixture of chromium green and titanium white pigments at fixed relative weight ratios (12–88% pigment ratio in the paint mixture). The average thickness of the applied layers varied between 193 and 257 μm, as measured by an analog profilometer (see Section 2). In this manner, we quantified the reduction of the PA imaging contrast as a function of paint thickness, resulting from both the intense optical diffusion and the higher ultrasonic attenuation. Furthermore, we additionally compared the contrast results obtained through PA imaging with the standard NIR method, evaluating their relative performance for different layers’ thickness. Figure 4a depicts the first sample covered by the thinnest paint layer, whereas Figure 4b corresponds to an image of the underlying pencil sketch prior to overpainting. The respective MAP PA image for an area of 3 by 3 cm^2^ is presented in Figure 4c, revealing the pencil sketch with an apparent high contrast. A NIR image of the same region is additionally shown in Figure 4d, providing information of the underdrawing with substantially lower contrast levels. Similar images of mock-ups covered by variable thickness paint layers of an identical constitution are presented in Figure 4e–h (average thickness: 220 μm), Figure 4i–l (average thickness: 229 μm), Figure 4m–p (average thickness: 247 μm), and finally Figure 4q–t (average thickness: 257 μm). The last two MAP PA and NIR images (Figure 4o,p,s,t) represent a slightly reduced scanning area of 2.5 by 2.5 cm^2^ to give more emphasis on the various spatial details located in the central region of the underlying sketch. It can be easily observed that in the cases of thicker paint layers, NIR imaging can hardly visualize the underlying pencil drawing, in contrast to respective PA reconstructions, which provide adequate contrast for this purpose. Following the completion of the imaging session, we proceeded to the contrast quantification of the recorded images, following a similar methodology to the one already described in Section 3.2.

Figure 5 shows a graph of the estimated contrast values as a function of the average paint layer thickness for PA and NIR imaging techniques. Error bars correspond to the standard error of the mean out of five measurements. The data points were fitted using simple linear models (red and blue lines) in the form *y = ax + b*, describing linear dependence of the imaging contrast reduction with an increasing layer thickness. The estimated fitting parameters (i.e., slope *a* and intercept point *b*), as well as the respective R^2^ values providing the goodness of fit measure, are explicitly presented in Table 2. The extracted results indicate that PA images provide a superior contrast over NIR images recorded at an excitation spectral band similar to the PA. This contrast improvement becomes more evident especially for higher thickness values and ranges between 4.1× for the thinner layer at 193 μm and 8.2× for the thickest layer at 257 μm, according to the predictions of the estimated linear models.

## 4. Discussion

In this study, we evaluated the performance of an epi-illumination photoacoustic apparatus for the detection of hidden underdrawings in specially designed mock-up samples that were covered by various paint layers composed of NIR transparent media and pigments such as gypsum, titanium white, minium, ultramarine blue, and chromium green. Moreover, we quantified the contrast of the PA images and compared our results with the standard optical NIR imaging technique. In this respect, experimental evidence demonstrates the better performance of the developed PA system in cases of highly scattering thick paint layers exceeding 250 μm in thickness, despite the narrowband signal detection (typically between 20 and 30 MHz) providing lower SNR values. Nevertheless, the main drawback of the proposed PA imaging approach is the necessity of an immersion medium (i.e., the distilled water), which is used to provide efficient acoustic coupling between the signal source and the detector, enabling also the transmission of the generated ultrasonic waves with minimal attenuation effects. Despite the fact that non-contact PA imaging of hidden layers in CH objects has not been demonstrated yet in an epi-illumination geometry, existing technologies and systems aiming at the contactless ultrasound detection could potentially record PA signals [28] with comparable sensitivity and resolution to conventional immersion piezoelectric transducers, requiring the application of a coupling medium. This capability was first demonstrated almost one decade ago [29] by using a two-wave mixing interferometer to inspect solid materials through the PA effect. More specifically, the emergence of high-performance polymer-based piezocomposite ultrasonic detectors supporting frequencies of a few MHz could provide non-contact PA signal detection in ambient air [30], as required ideally by the proposed diagnostic method. Furthermore, optical detection of ultrasound-based on refractometric or interferometric techniques has an experimentally proven potential on the non-contact recording of broadband PA signals, with bandwidths that are similar to conventional piezoelectric elements [31,32]. Consequently, such examples of novel technologies could be integrated into a future upgraded version of the reflection-mode PA imaging system, which will be able to offer highly specific information with sufficient contrast and spatial resolution but without the need for any contact with the investigated surfaces. Apart from this important improvement, a multispectral PA excitation through tunable laser sources could additionally enable the differentiation of several hidden layers demonstrating variable optical absorption properties. Under such conditions, optimized spectral unmixing algorithms [33] may provide quantitative information on the relative local concentration of each pigment, uncovering also the full stratigraphy of the investigated object with high resolution. Finally, the total image acquisition time could be dramatically reduced from a couple of hours to several minutes by integrating Q-switched nanosecond lasers operating at higher pulse repetition rates. The integration of such technological improvements could make the proposed imaging method a powerful tool for the in-depth investigation of various artworks, including wall paintings, frescos, sculptures, or even documents [34], significantly complementing the existing diagnostic approaches [35] for this purpose.

## 5. Conclusions and Future Perspectives

This work demonstrated the capabilities and the potential of a novel reflection-mode PA imaging system for the detection of hidden underdrawings in specially designed multilayered artwork samples. The system is proven to overcome the limitations of previous transmission geometry setups, opening a path to investigations on a wide range of cultural heritage objects with diverse geometrical and structural features. The obtained results prove that PA signals reveal underlying sketches with up to 8 times improved contrast, as compared to the traditional NIR optical technique. Therefore, we demonstrated a strong advantage of the novel epi-illumination PA imaging setup for a wide range of cultural field applications. The main challenge to be faced in future implementations is the non-contact and non-invasive recording of the PA signals. Further developments, combined with the demonstrated performance, are expected to substantially expand and improve the applicability of the PA imaging technique in heritage science.

## Figures and Tables

**Figure 1 jimaging-07-00183-f001:**
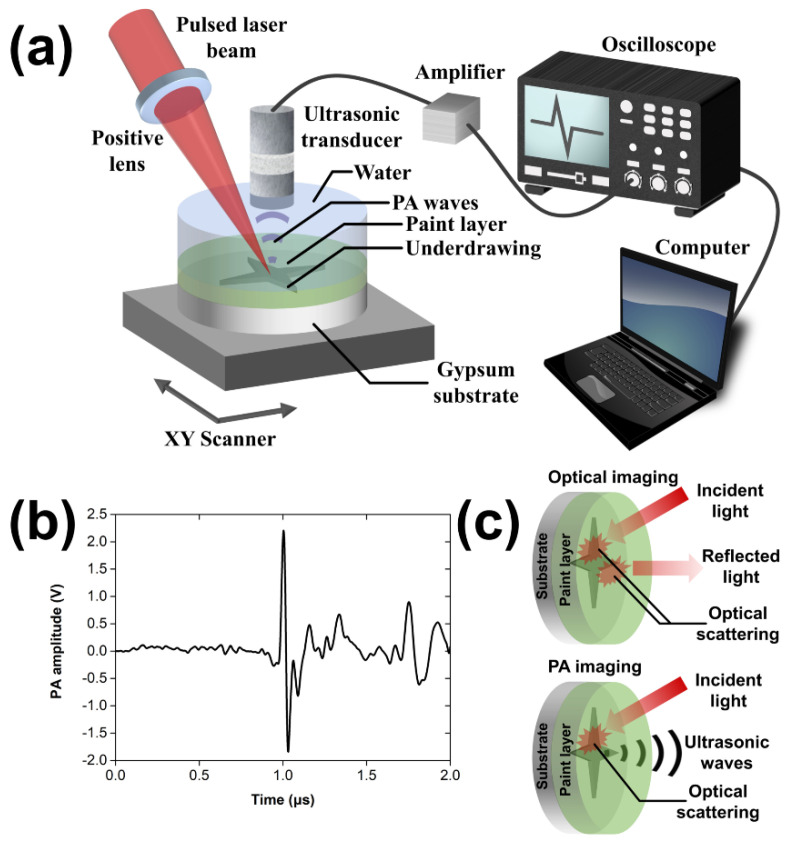
(**a**) Three-dimensional scheme of the reflection-mode photoacoustic (PA) imaging apparatus; (**b**) typical PA signal recorded in the time domain; (**c**) principle of optical and PA imaging for the detection of hidden underdrawings.

**Figure 2 jimaging-07-00183-f002:**
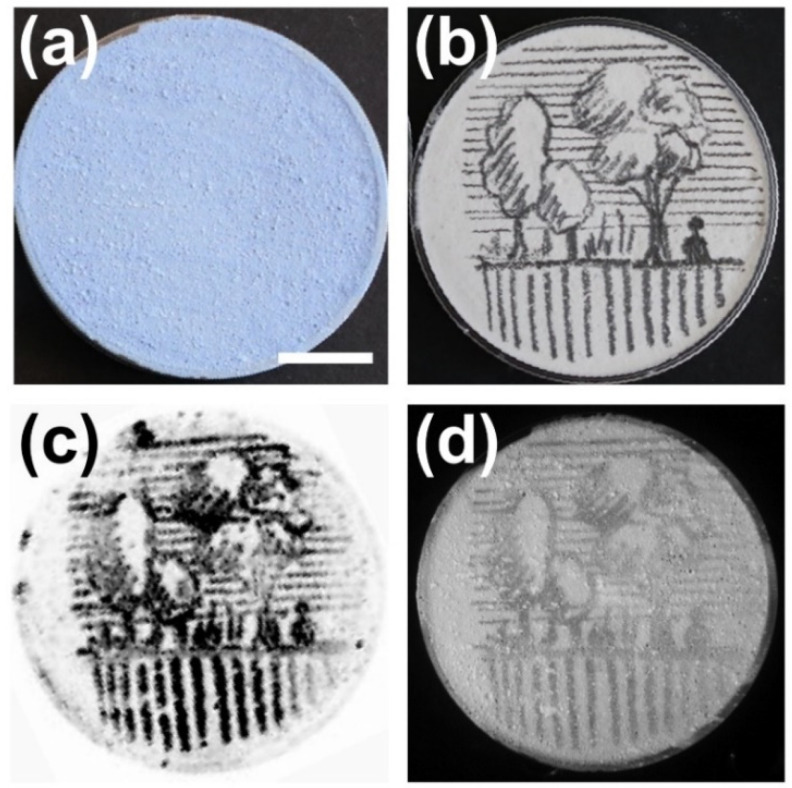
Images (**a**) of a mock-up covered with a paint layer containing a mixture of titanium white, gypsum, and ultramarine blue pigments, (**b**) of an underlying pencil sketch prior to the application of the paint; (**c**) maximum amplitude projection (MAP) PA reconstruction of the hidden underdrawing; (**d**) respective near-infrared (NIR) image recorded at 1100 nm. Scale bar corresponds to 1 cm.

**Figure 3 jimaging-07-00183-f003:**
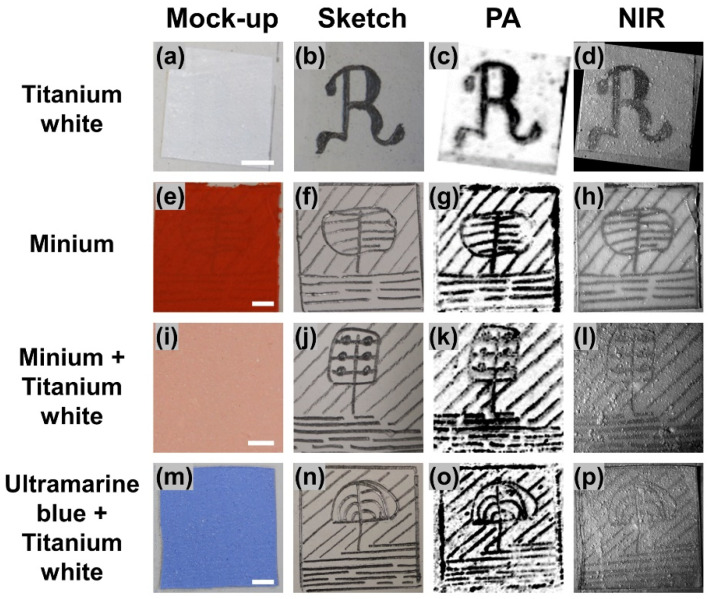
Images (**a**) of a mock-up covered by a titanium white paint layer, (**b**) of the underlying pencil sketch prior to overpainting; (**c**) MAP PA reconstruction of the hidden letter “R”; (**d**) respective NIR image recorded at 1100 nm. Analog results are presented for minium (**e**–**h**), minium plus titanium white (**i**–**l**), as well as ultramarine blue plus titanium white paint layers (**m**–**p**). All scale bars are equal to 5 mm.

**Figure 4 jimaging-07-00183-f004:**
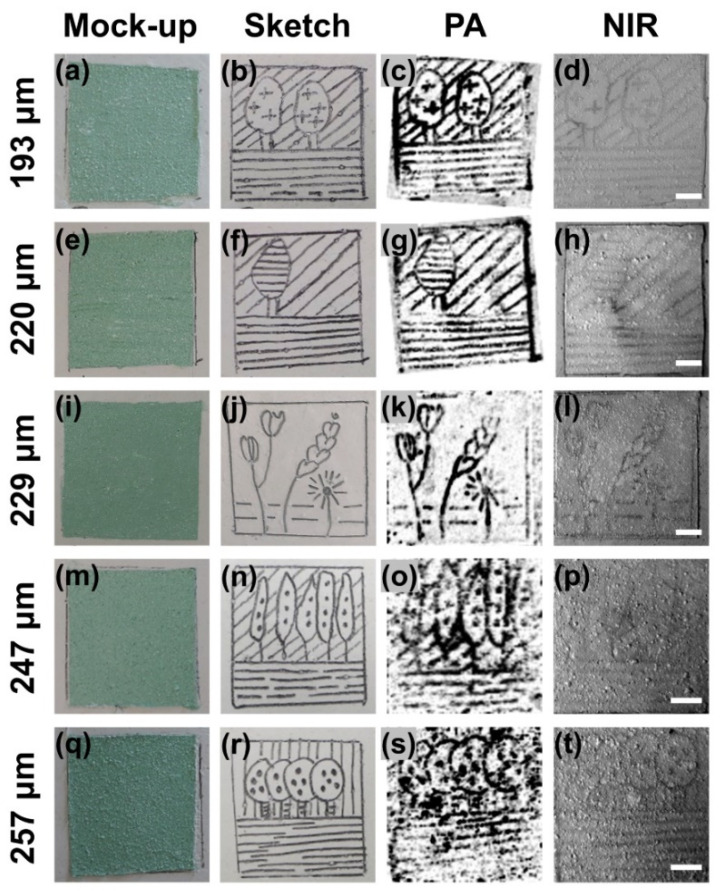
Images (**a**) of a mock-up covered by a mixture of chromium green and titanium white paint layer, (**b**) of the underlying pencil sketch prior to overpainting; (**c**) MAP PA reconstruction of the hidden sketch; (**d**) respective NIR image recorded at 1100 nm. Similar results are presented for mock-ups covered by paint layers of identical composition but with a gradually increasing average thickness, as presented on the left side of the panel (**e**–**t**). All scale bars are equal to 5 mm.

**Figure 5 jimaging-07-00183-f005:**
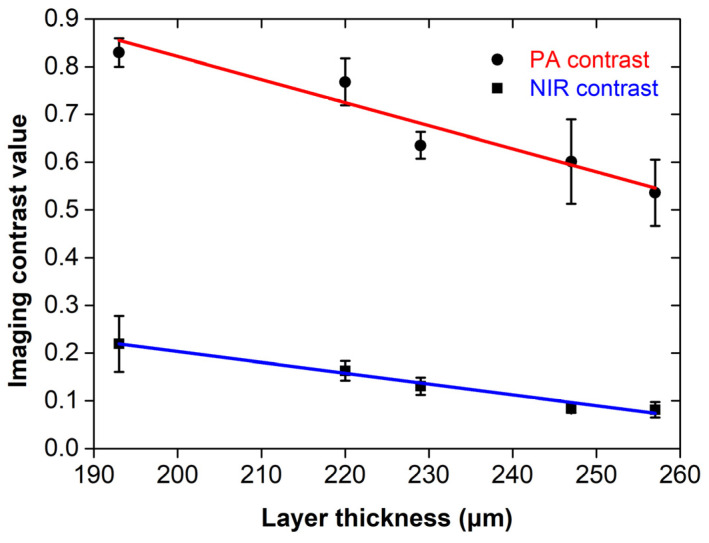
Plot of imaging contrast values for PA (black circles) and NIR (black squares) techniques as a function of the average paint layer’s thickness. Error bars represent the standard error of the mean out of five measurements. Red and blue lines show the linear fitting of the respective data points.

**Table 1 jimaging-07-00183-t001:** Contrast quantification for PA (C_PA_) and NIR optical images (C_NIR_) of mock-ups covered by different pigmented layers (thickness 250 μm). Uncertainties correspond to the standard error of each mean out of five measurements.

Pigment	C_PA_	C_NIR_	C_PA_/C_NIR_
Titanium white	0.94 ± 0.01	0.27 ± 0.04	3.5
Minium	0.80 ± 0.10	0.21 ± 0.03	3.8
Minium + titanium white	0.91 ± 0.02	0.30 ± 0.05	3.0
Ultramarine blue + titanium white	0.89 ± 0.03	0.13 ± 0.01	6.8

**Table 2 jimaging-07-00183-t002:** Fitting parameters for the linear models (*y = ax + b*) presented in Figure 5, describing the imaging contrast reduction as a function of the overlying layer’s thickness.

Parameter	PA	NIR
Slope (*a*)	−4.8 × 10^−3^ ± 7 × 10^−4^	−2.3 × 10^−3^ ± 1 × 10^−4^
Intercept (*b*)	1.80 ± 0.20	0.66 ± 0.03
R^2^	0.912	0.986

## Data Availability

The data that support the findings of this study are available from the corresponding authors, [G.J.T and A.C.], upon reasonable request.

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
