# Peer review of "Revealing Hidden Features in Multilayered Artworks by Means of an Epi-Illumination Photoacoustic Imaging System"

_2313-433X, 2021, doi:10.3390/jimaging7090183_

Round 1
Reviewer 1 Report
The manuscript (MS) entitled as "Revealing hidden features in multilayered artworks by means of an epi-illumination photoacoustic imaging system" by G. Tserevelakis at al. tells about the application of photoacoustic microscopy in art conservation and evaluation. The the detection of the signal in back reflection direction (epi detection) which is described in this work is rather new in the field with clearly stated advantages which might be of the potential interest and significant impact. The MS is logically organized, understandable with clearly presented results. There are some minor, concerns that should be addressed before it can be considered for publishing in this journal.
1.The epi detection in photoacoustic is rather rare and applied only in biomedical imaging
https://www.spiedigitallibrary.org/journals/journal-of-biomedical-optics/volume-18/issue-03/030501/Reflection-mode-multifocal-optical-resolution-photoacoustic-microscopy/10.1117/1.JBO.18.3.030501.full?SSO=1 https://www.nature.com/articles/s41598-020-76155-6 https://www.sciencedirect.com/science/article/pii/S2213597914000330 https://arxiv.org/ftp/arxiv/papers/2103/2103.17035.pdf .
Could authors give a short comment what and why the epi detection in photoacoustic microscopy for artwork studies is different from that one for biomedical studies (described in the papers stated above). Could the main characteristics of both be emphasized and compared. Also, from the abstract the readers might have impression that the epi detection in the phtocoustic microscopy presented in this MS is something exclusively new in general, in any filed. Could authors correct this and limit their contribution in epi detection for cultural heritage diagnostics.
2.Fig 2a, 3a,ei,m and 4a,e,i,m,q are denoted as bright field images. The bright field images are usually met at transmitted light microscopy where the sample is rather thin in which the inhomogeneities contributing the light extinction give rise to the contrast and where the homogenous bright filed is seen in the absence of the sample (hence the name "bright field). From my experience and taking into account that the samples studied here are tick and opaque I would say that the aforementioned images might be rather dark filed meaning recorded in the back reflection geometry where the uniform dark field is seen in the absence of the sample. Could authors pay attention to this and correct the figure captions and the body text accordingly or give the comment if I am wrong.
3.Results presented in Fig 5 show linear dependence of the contrast on the sample thickness. Having in mind that the intensity of the light does not depend linearly on the length (e.g. Lambert Beer law) one would naively intuitively expect nonlinear dependence in Fig 5. Could the authors give the comment on this issue?
4.Since authors mentioned several times that the presented method is used to visualize "hidden" features, could it (or photoacoustic microscopy in general) be used for document security and/or forensic applications? For instance, could the PA (microscopy) can be used for machine (2nd inspection line) and/or forensic (3rd inspection line) visualization of deliberately covert features in order to check their authenticity e.g. to figure out if they are counterfeited. Could authors comment on this, provide the references if there are any and compare to other imaging/microscopy techniques that are already used in this field.
Reviewer 2 Report
The authors presented a standard photoacoustic microscopy (PAM) set-up (Fig. 1) using a piezoelectric focusing acoustic transducer with 73 MHz center frequency which needs water coupling to reveal hidden underdrawings from a graphite pencil. The signal was bandpassed between 100 kHz and 30 MHz to estimate the peak-to-peak PAM amplitude. The PAM image was acquired by scanning a 300 micron raster and was compared to a standard high-resolution CMOS camara image coupled with a NIR-transparent objective lens. The contrast of the PAM and the NIR images was quantified and compared and experimental evidence demonstrates better performance of the PAM system, but with the need of a coupling-media, such as water.
In general:
As mentioned in Section 4, “the main challenge… is the non-contact and non-invasive recording of the PA signals, which could substantially expand and improve the applicability of the imaging method”. Such non-contact detection systems already exist! Not only for laser ultrasonic detection but also for photoacoustic imaging, see a review article in Photoacoustics [Z. Hosseinaee et al, Towards non-contact PA imaging, Volume 20]. To our knowledge, the first work on remote photoacoustic imaging on solid material was already published in 2010 by T.Berer et al, Optics Letters 35, which was cited in the meantime by 50 other papers on non-contact PA imaging. Please give adequate references and mention this in your manuscript.
In detail:
Ad Section 3.1 “PA imaging setup” should be section 2.1 – wrong numbers (also in 3.2, 3.3 and 3.4): Why did you use a 73 MHz center frequency transducer and then cut at 30 MHz? For that frequency band a center frequency around 15 MHz would have been better for a higher SNR. Please describe this in the text.
Ad Section 3.4 “NIR imaging”: what is pixel size with your NIR imaging setup compared to the 300 microns pixel size of PAM? Please give this for comparison.
Reviewer 3 Report
Authors developed an epi-illumination photoacoustic apparatus for diagnosis in heritage science. The manuscript is written well. However, there is one major comment. In PA imaging, CR or CNR are not valid any more. Author need to calculate gCNR for contrast evaluation. See :
Kempski KM, Graham MT, Gubbi MR, Palmer T, Bell MA. Application of the generalized contrast-to-noise ratio to assess photoacoustic image quality. Biomedical Optics Express. 2020 Jul 1;11(7):3684-98.
Rodriguez-Molares A, Rindal OM, D’hooge J, Måsøy SE, Austeng A, Bell MA, Torp H. The generalized contrast-to-noise ratio: a formal definition for lesion detectability. IEEE Transactions on Ultrasonics, Ferroelectrics, and Frequency Control. 2019 Nov 29;67(4):745-59.
Round 2
Reviewer 1 Report
the manuscript has been improved upon the revision and it can be published in J Imag.
Reviewer 2 Report
The authors have taken into consideration all of the comments and have proceeded to the required changes in the text.
Author Response
We would like to thank once more the Reviewer for his valuable comments.
Reviewer 3 Report
My commnet is addressed now.